# An Organic–Inorganic Hybrid Nanocomposite as a Potential New Biological Agent

**DOI:** 10.3390/nano10122551

**Published:** 2020-12-18

**Authors:** Mateusz Dulski, Katarzyna Malarz, Michał Kuczak, Karolina Dudek, Krzysztof Matus, Sławomir Sułowicz, Anna Mrozek-Wilczkiewicz, Anna Nowak

**Affiliations:** 1Institute of Materials Engineering, University of Silesia, 75 Pulku Piechoty 1a, 41-500 Chorzow, Poland; 2Silesian Center for Education and Interdisciplinary Research, 75 Pulku Piechoty 1a, 41-500 Chorzow, Poland; katarzyna.malarz@us.edu.pl (K.M.); mkuczak@us.edu.pl (M.K.); anna.mrozek-wilczkiewicz@us.edu.pl (A.M.-W.); 3A. Chełkowski Institute of Physics, University of Silesia, 75 Pulku Piechoty 1, 41-500 Chorzow, Poland; 4Institute of Chemistry, University of Silesia, Szkolna 9, 40-007 Katowice, Poland; 5Łukasiewicz Research Network - Institute of Ceramics and Building Materials, Refractory Materials Division in Gliwice, Toszecka 99, 44-100 Gliwice, Poland; karolina.dudek@icimb.lukasiewicz.gov.pl; 6Materials Research Laboratory, Silesian University of Technology, Konarskiego 18a, 44-100 Gliwice, Poland; krzysztof.matus@polsl.pl; 7Institute of Biology, Biotechnology and Environmental Protection, University of Silesia, Jagiellonska 28, 40-032 Katowice, Poland; slawomir.sulowicz@us.edu.pl; 8Institute of Nuclear Physics Polish Academy of Sciences, PL-31342 Krakow, Poland; ana.maria.nowak@gmail.com

**Keywords:** chemical reduction, silver-silica nanocomposite, carboxymethylcellulose, sodium alginate, physicochemical features, antimicrobial activity, anticancer activity, colon cancer, breast cancer, pancreatic cancer, glioblastoma

## Abstract

To solve the problem of human diseases caused by a combination of genetic and environmental factors or by microorganisms, intense research to find completely new materials is required. One of the promising systems in this area is the silver-silica nanocomposites and their derivatives. Hence, silver and silver oxide nanoparticles that were homogeneously distributed within a silica carrier were fabricated. Their average size was d = (7.8 ± 0.3) nm. The organic polymers (carboxymethylcellulose (CMC) and sodium alginate (AS)) were added to improve the biological features of the nanocomposite. The first system was prepared as a silver chlorine salt combination that was immersed on a silica carrier with coagulated particles whose size was d = (44.1 ± 2.3) nm, which coexisted with metallic silver. The second system obtained was synergistically interacted metallic and oxidized silver nanoparticles that were distributed on a structurally defective silica network. Their average size was d = (6.6 ± 0.7) nm. Physicochemical and biological experiments showed that the tiny silver nanoparticles in Ag/SiO_2_ and Ag/SiO_2_@AS inhibited *E. coli*, *P. aeruginosa*, *S. aureus*, and *L. plantarum’s* cell growth as well as caused a high anticancer effect. On the other hand, the massive silver nanoparticles of Ag/SiO_2_@CMC had a weaker antimicrobial effect, although they highly interacted against PANC-1. They also generated reactive oxygen species (ROS) as well as the induction of apoptosis via the p53-independent mechanism.

## 1. Introduction

Progress in biomedicine is closely related to the still-developing nanotechnology. In this context, nanotechnology enables novel multifunctional materials to be fabricated that contribute to improving everyday life [1], e.g., engineering new medicines in combination with porous materials/carriers or well-known organic or inorganic materials to be scaled-down into the “nano” range. Both approaches have advantages and disadvantages because the first route is designed to increase the drug selectivity of the action, while the second route may cause a higher dispersion of the material, which might translate into an increase in clinical effectiveness [2,3]. Hence, nanotechnology is expected to offer the possibility to change the form of the materials that are already being used and improve their functionality and properties [4].

Another route for improving the biological features of systems using a molecularly desired structure is its combination with polysaccharides, which have been explored in their natural state in nanomedicines for many years [5]. In this context, polysaccharides are classified into three types according to their origin: vegan (e.g., starch, cellulose, pectin), animal (e.g., chitin, heparin), and microbial/fungal (e.g., dextran. alginate) [5], wherein all of them can be considered to be carriers, especially in injectable drug delivery systems. Unfortunately, in this field, their stabilizing properties are still a challenge. Therefore, new methods for utilizing polysaccharides, especially for improving their physicochemical properties as well as their biomedical potential, are still being sought. Alginates, including sodium alginate (AS), which are cell-wall constituents of brown algae or a cellulose derivative such as carboxymethylcellulose (CMC), are increasingly common in various industries. Sodium alginate can be easily gelled in an acid medium. Due to its high similarity to extracellular matrices of living tissues, it can prepare materials to heal wounds or deliver bioactive agents such as drug proteins. AS can also be used to encapsulate materials, which causes the slow-release of a metallic nanostructure [6] or drugs [7] into cells in a controlled manner. Hence, alginate-based hydrogels or composite derivatives can be systems that can potentially be used to deliver nanostructures into the interior of cells [8,9]. It can also be used to help cure gastroesophageal reflux in children [10]. In turn, its non-antigenicity or mucoadhesiveness makes AS an ideal material for creating dental impressions, wound dressings, or engineering tissue [11]. In turn, CMC is widely used in biomedical applications and detergents, textiles, paper, food, and oil exploration [12]. It is also used as an emulsifying agent to improve a system’s viscosity and prevent material from settling when being stored [13]. It has well-wetting and valuable features for particle coagulating or gluing due to its excellent film-forming properties. CMC’s adhesive features could be useful for improving bonding particles to a base or carrier during the manufacturing process.

Other papers have also reported that polysaccharides can be used as a carrier for metallic nanoparticles, improving wound healing on the one hand or acting as a biological agent on the other [14]. The latter application, especially within developing new types of anticancer materials, has aroused the broadest interest. Cancer is a multifaceted disease caused by a complex combination of genetic and environmental factors that require a multistage treatment [15]. There are several therapeutic methods, and many of them are focused on using therapies that affect the entire body, not only the pathogenically changed site. Targeted treatments seem to be one of the most desirable solutions, and silver nanoparticles (Ag NPs) or silver-based hybrid systems can be considered one of the potentially new therapeutic agents. According to the literature, Ag NPs can cause cytotoxicity in cancer cells through various mechanism that involve oxidative stress [16,17,18,19], DNA damage [20], cell cycle arrest [17,19,21,22], apoptosis and necrosis [16,23,24]. Lin et al. demonstrated that silver nanoparticles could induce cytoprotective autophagy, which improves the efficacy of Ag NPs in anticancer therapy [25].

Unfortunately, the lack of understanding of nanotoxicology and the biological activity of silver nanoparticles limits their usefulness. These might be associated with their aging process [26], thermal stability [27] or oxidation [28,29], as well as their shape, size [30,31,32,33,34], morphology [35,36,37], or degree of silver agglomeration [34,38,39]. Therefore, using a combination of Ag NPs with polysaccharides seems to be an idea that could reduce the number of silver ions released into the environment and reduce their toxicity. Augustine and Rajarathinam showed that a combination of Ag NPs with alginate reduced the toxicity of the silver ions released from silver nanoparticles and improved wound healing [40]. The same effect was observed in CMC by Kumar et al. [41] and Prema et al. [42]. Unfortunately, there is a growing bacterial resistance to silver, which is alarming when searching for new antibiotics. Therefore, using polysaccharides seems to be an interesting direction that might stabilize the silver-based systems. Unfortunately, some negative aspects of such an approach to the organic–inorganic system were revealed by Lengert et al., who showed that the influence of ultrasounds influences nanoparticles’ release from such a capsule [43]. Hence, another carrier must improve the stability of silver and ensure its slow release from the matrix.

Recently, there have been many papers in the literature that have highlighted an innovative approach for functionalizing well-known structures by combining porous materials such as an oxide matrix carrier, e.g., a polymer, silicon oxide, titanium oxide, or zinc oxide, with the ionic form of metals or with metallic nanoparticles [44,45,46,47]. The chemically modified fumed silica seems to be the most promising inorganic structure due to the immobilization of silver on the surface via unsaturated chemical bonds [48]. Another advantage of such a molecular combination is that the unsaturated -Si-O* bonds, which is the biological factor, improve the response of the bioactive materials at their interface [49]. However, the effectiveness of the biological interaction of Ag/SiO_2_ depends strongly on the type of reagents and their concentration in a solvent, chemical purity (>99.999%), stabilizing agents (glycerin, polyethylene glycol, organic polymers, etc.), physical conditions (temperature, precursor, reactant concentration, microwave field, etc.) [50], and the manufacturing procedure that is used such as chemical reactions [50,51,52], sol-gel [53], or sputtering deposition [54]. These factors and methods provide an opportunity to control the physical parameters of a chemical reaction. An interesting approach to fabricating hybrid structures could also be modifying a chemical procedure by using polysaccharides to stabilize the silver or even encapsulate a silver-silica composite to improve the initial material’s biological features. 

Therefore, this paper illustrates a new approach to developing bioactive modified silver-silica structures, especially in the chemical modification of such molecular structures using the commercially available polysaccharides (carboxymethylcellulose and sodium alginate). The impact of all of the components on the shape, size, and morphology of a particle as well as its phase composition was determined using X-ray diffraction (XRD), scanning (SEM), and transmission (TEM) electron microscopy. The molecular interactions and structural parameters were investigated using Raman spectroscopy. The electronic structure and oxidation state of the silver were determined using X-ray photoelectron spectroscopy (XPS). Lastly, the structural and chemical parameters were correlated with the antimicrobial activity as well as the cytotoxicity on a group of cancer cells that included colon (HCT116), breast (MCF-7), pancreatic (PANC-1), and brain (U-251) cancers. The synthesized nanocomposites’ impact on inhibiting the cell cycle and inducing apoptosis by generating reactive oxygen species (ROS) in the pancreatic cells was also examined. Therefore, the developed nanocomposites have the potential to be new, biologically active agents.

## 2. Materials and Methods

### 2.1. Synthesis

All of the primary chemical ingredients such as the silver nitrate, sodium hydroxide (purity above 99.99%; Avantor Company, Gliwice, Poland), amorphous silica (Orisil 380; Orisil Ltd., Lviv Ukraine), and technical grade organic polymers such as carboxymethylcellulose (CMC) and sodium alginate (AS) were used. The technical CMC was a mixture of sodium chloride and sodium glycolate, while AS contained a residual fraction of sodium hypochlorite or chlorine dioxide. Distilled water was used to prepare all of the samples.

The silver-silica nanocomposite synthesis and its derivatives were carried out in two stages (Figure 1). The first step was preparing the carrier, i.e., 10 g silicon dioxide mixed with 200 mL of water. Next, the aqueous silica solution was subjected to microwave irradiation for 120 s to activate the silica’s surface. A 4% aqueous sodium hydroxide solution and then silver nitrate was added dropwise into the silica suspension in the final step. The silica suspension was continuously mixed at room temperature on a magnetic stirrer for 1 h. The result was a silver-silica colloid. The final product was filtered on a polyethylene filter, washed, and dried at room temperature (route A in Figure 1).

A similar procedure was used to fabricate the hybrid organic–inorganic systems. The silver-silica colloid was prepared as before. However, a 4% aqueous CMC or AS solution was added dropwise to the suspension to encapsulate the silver-silica system and form a core-shell organic–inorganic composite (route B in Figure 1). Lastly, both samples were filtered twice on a polyethylene filter, washed, and dried at room temperature to remove any impurities due to the synthesis.

### 2.2. Experimental Methods

An X’PertPro MPD PANalytical (Malvern PANalytical, Almelo, The Netherlands) X-ray diffractometer with a Cu *Kα* radiation (λ = 1.54 Å) was used to obtain the crystal structure of the Ag/SiO_2_ composite. The lattice parameters were refined using the Rietveld method with HighScore Plus (version 4.9) software (Malvern PANalytical, Almelo, The Netherlands) and the ICCD PDF-4+ database. The crystallite size was analyzed based on the change of the profile widths compared to a standard sample according to the Scherrer equation in HighScore Plus version 4.8. Calcite was used as the standard sample.

Particle morphology was determined using Scanning (SEM) and Transmission (TEM) Electron Microscopy with energy dispersive spectrometer (EDS). By contrast, the individual elements’ surface chemical distribution was studied using a scenning electron microscopy SEM and X-ray photoelectron spectroscopy (XPS). The scanning electron microscopy (SEM) data were obtained using a TESCAN Mira 3 LMU equipped with an energy dispersive spectrometer (EDS) (TESCAN, Brno, Czech Republic). Images were collected with secondary electron (SE) and backscattered electron (BSE) detectors. Samples covered with a carbon layer were measured using Quorum Q150T Equipment (United Kingdom) to compensate for the charge of the system. The TEM micrographs were collected using a probe C_s_-corrected S/TEM Titan 80–300 FEI (FEI Company, Hillsboro, OR, USA) microscope equipped with an EDAX EDS detector. The images were recorded in high resolution scanning transmission electron microscopy (HRSTEM)-mode using a high-angle annular dark-field (HAADF) detector. The XPS measurements were collected using a Prevac and VGScienta spectrometer (Newburyport, MA, USA). Monochromatic Al *K_α_* radiation (1486.6 eV) was used to collect the broad energy range survey spectra as well as the spectra of the core levels, including Ag *3d*, O *1s*, Si *2p*, and Cl *2p*. The chemical composition and oxidation states within the initial Ag/SiO_2_ and its modification were also determined. The spectra were shifted relative to the carbon C *1s* line at 284.6 eV and corrected for the background signal using the integrated Shirley algorithm. The bands were fitted by composing the Gaussian and Lorentzian lines using MultiPak (version 9.6.1.) software.

Structural analysis, especially characterizing the three-dimensional network of the silica and identifying the individual phases, was performed using Raman spectroscopy. A WITec Confocal Raman Microscope (CRM) alpha 300 R equipped with an air-cooled solid-state laser (λ = 532 nm) and a charge-coupled device (CCD) camera was used. The excitation laser radiation was coupled with a microscope using a single-mode optical fiber with a 50 nm diameter. An air Olympus MPLAN (100×/0.90NA) objective was used during the measurements. The Raman scattered light was focused onto a multi-mode fiber (100 nm diameter) and a monochromator with a 600 line/mm grating. The Raman spectra were accumulated in 100 scans at an integration time of 20 s and a resolution of 3 cm^−1^. The spectrometer monochromator was calibrated using the Raman scattering line of a silicon plate (520.7 cm^−1^). The post-processing analysis, with the baseline correction and cosmic ray, was performed using WITecProjectFour (version 4.1) Plus software. The peak fitting was analyzed using the GRAMS (version 9.2) software package.

### 2.3. Antimicrobial Properties of Nanocomposites—Spot Test

The toxicity of Ag/SiO_2_ and its modified derivatives, Ag^+^-ions, was examined against the microbial stains Gram-negative bacteria *Escherichia coli* and *Pseudomonas aeruginosa*, Gram-positive bacteria *Lactobacillus plantarum* and *Staphylococcus aureus*, and the yeast *Candida albicans* and *Saccharomyces cerevisiae*. The cell suspensions were prepared as was previously described [54,55]. Cell suspensions of 24 h bacterial and yeast culture (Luria Broth (LB) and and Yeast extract Peptone Dextrose (YPD) media, respectively, at 301 K, 160 rpm) were diluted in fresh growth medium and cultivated (301 K, 160 rpm) for four to six hours. When the mid-exponential growth phase for the bacteria (OD_600nm_ = 0.6) and yeast (OD_600nm_ = 0.8–1.0) was reached, the cell suspensions were centrifuged (4700 rpm for 10 min at 293 K), the pellets were suspended in sterile demineralized (DI) water (Milli-Q, Millipore, Darmstadt, Germany) and centrifuged again. Finally, the cells were resuspended in DI water to a density of ~10^7^ CFU/mL for the bacteria (OD_600nm_ = 0.1) and yeast (OD_600nm_ = 1.2) as was described by Suppi et al. [54].

Next, 30 µL of the cell suspension was added to the 30 µL of samples and ion forms of Ag^+^ (AgNO_3_), treated as a reference. All of the samples were tested at six nominal concentrations: 0.01, 0.1, 1.0, 10, 100, and 1000 mg/L. The stock solutions (2000 mg/L) of the nanomaterials and Ag^+^ were placed in sterile DI water and homogenized using an ultrasonic probe (PolSonic 3, Warsaw, Poland) at 2 × 160 W for 1 h immediately before the test. The stocks were diluted in DI water and the tested concentrations were distributed into 96-well microplates (BD Falcon). The test organisms were exposed to the samples in DI water at 297 K for 24 h without shaking in the dark. Each experiment was repeated three times.

After a 24 h exposure, 4 µL of the cell suspension was pipetted as a ‘spot’ onto LB or YPD agar plates, which were incubated for 24 h (bacteria) or 72 h (yeast) at 301 K. The formation of a visible ‘spot’ (colonies) was used to determine the ability of the tested organisms to grow on an agar medium. The minimum biocidal concentration (MBC) of the tested nanomaterials and ion forms of silver was determined as the lowest tested concentration of a chemical that completely inhibited the formation of visible colonies after subculturing on toxicant-free agar media.

### 2.4. Anticancer Studies

#### 2.4.1. Cell Culture

The human colon carcinoma cell line HCT 116 and the human breast carcinoma cell line MCF-7 were obtained from ATCC. The human glioblastoma cell line U-251 was provided by the Institute of Cancer Research in London, England. The human pancreatic carcinoma cell line PANC-1 was purchased from Sigma Aldrich (Saint Luis, MO, USA). The cells were grown as monolayer cultures in 75 cm^2^ flasks (Nunc) in Dulbecco’s modified Eagle’s medium (DMEM). The DMEM was supplemented with 12% heat-inactivated fetal bovine serum (Sigma, Saint Luis, MO, USA) and 1% *v*/*v* of penicillin/streptomycin (Gibco). The cells were cultured under standard conditions at 311 K in a humidified atmosphere at 5% CO_2_. All of the cell lines were subjected to routine mycoplasma testing using the PCR technique with specific *Mycoplasma* primers to ensure no contamination.

#### 2.4.2. Cytotoxicity Studies

The tested nanocomposites, Ag/SiO_2_, Ag/SiO_2_@CMC, and Ag/SiO_2_@AS, were dissolved in the culture medium to achieve the necessary concentrations. The exponentially growing cells were harvested by trypsinizing the sub-confluent cultures. The cells (HCT 116, MCF-7, U-251, PANC-1) were seeded into 96-well cell culture microtiter plates (Nunc, Roskilde, Denmark) at concentrations of 5000 cells per well and incubated at 310 K for 24 h. After this time, the growth medium was exchanged for a medium containing the tested nanocomposites at concentrations ranging from 1 to 50 mg/L. The cells were incubated with the nanocomposites for 72 h under standard cell culture conditions. Then, the medium was replaced with 100 µL of DMEM without phenol red. The viable cells’ metabolic activity was determined by adding 20 µL of CellTiter 96AQueousOne Solutions—MTS (Promega, Medison, WI, USA) and followed by a 1 h incubation. The MTS assay is a colorimetric method that is used to determine the number of viable cells. Here, a standard solution containing 100 µL of DMEM without phenol red and 20 µL of an MTS solution was used to determine the “blank” absorbance. By contrast, the absorbance of Ag/SiO_2_ was measured at 490 nm using a Synergy™ 4 microplate reader (BioTek, Winooski, VT, USA). A 50% inhibitory concentration (IC_50_) was defined as the compounds’ concentration that can reduce cell proliferation to 50% of the untreated control cells. Each compound was individually tested in triplicate in a single experiment, while each experiment was repeated four times. The IC_50_ values were calculated using GraphPad Prism (verion 7) software.

#### 2.4.3. Time-Dependent Measurement of the ROS Level

The PANC-1 cells were seeded into black 96-well plates (Corning, Corning, NY, USA) at a density of 9000 cells/well and incubated at 310 K. After incubating overnight, the tested nanocomposite Ag/SiO_2_@CMC (30 mg/L) solution was added and then incubated for 1, 3, 6, 9, 12, and 24 h in a kinetic experiment. The generation of ROS was measured using a CellROX^®^ Green Reagent (Molecular Probes™, Eugene, OR, USA). Additionally, the number of cells in each well was determined using Hoechst 33342 (Molecular Probes™). The tested compounds’ solutions were removed, and 100 µL of CellROX Green Reagent and Hoechst 33342 at a final concentration of 5 µM were added to each well. Then, the cells were incubated for 30 min at 310 K. The fluorescence was measured using a multi-plate reader (Synergy 4, Bio Tek, Winooski, VT, USA) at 485 nm excitation and a 520 nm emission for CellROX Green Reagent and a 345 nm excitation laser and a 485 nm emission filter for the Hoechst 33342. The experiments were each performed three to four times. The ROS levels are expressed as the percentage of the level of the control cells.

#### 2.4.4. Cell Cycle Assay

The PANC-1 cells were seeded in 3 cm Petri dishes (Nunc) at a density of 0.25·10^6^ cells/well and incubated at 310 K for 24 h. The medium was then removed, and freshly prepared solutions of the tested nanocomposites, Ag/SiO_2_, Ag/SiO_2_@CMC, and Ag/SiO_2_@AS were added at a 30 mg/L concentration. After a 48 h treatment, the assay was performed using a Muse Cell-Cycle Kit (Millipore, Burlington, MA, USA) according to the manufacturer’s instructions. The cells were collected, washed with cold phosphate-buffered saline (PBS), and centrifuged at 300× *g* for 5 min. Then, the cells were fixed in ice-cold 70% ethanol and stored at 253 K overnight. Afterward, the cells were centrifuged, resuspended in 200 µL of Muse™ Cell Cycle Reagent, and incubated for 30 min at room temperature in the dark. After staining, the cells were processed for cell cycle analysis using a Muse Cell Analyzer (Millipore, Burlington, MA, USA). The experiments were performed at least three times.

#### 2.4.5. Annexin V Binding Assay

The PANC-1 cells were seeded in 3 cm Petri dishes (Nunc, Roskilde, Denmark) at a density of 0.25·10^6^ cells/well and incubated at 310 K for 24 h. The medium was then removed, and freshly prepared solutions of the tested nanocomposites, Ag/SiO_2_, Ag/SiO_2_@CMC, and Ag/SiO_2_@AS, were added at a 30 mg/L concentration. After 48 h, the assays were performed using an Annexin V and Dead Cell Kit (Millipore, Burlington, MA, USA) according to the manufacturer’s instructions. Briefly, detached and adherent cells were collected and centrifuged at 500× *g* for 5 min. Next, the resuspended cells were incubated with 100 µL of Muse™ Annexin V & Dead Cell Reagent for 20 min at room temperature in the dark. After staining, the events for life, early and late apoptotic cells were counted using a Muse Cell Analyzer (Millipore, Burlington, MA, USA). The experiments were performed at least three times.

#### 2.4.6. Immunoblotting

The PANC-1 cells were seeded in 3 cm Petri dishes (Nunc) at a density of 0.5·10^6^ cells/well and incubated overnight. The next day, solutions of the nanocomposite Ag/SiO_2_@CMC (30 mg/L) and DOX (5 µM concentration) were added, and the cells were incubated for 24 h. Cells were harvested via trypsinization and washed with cold PBS. Next, the cells were centrifuged and suspended in a RIPA buffer (Thermo Scientific, Waltham, MA, USA) containing a Halt Protease Inhibitor Cocktail (Thermo Scientific, Waltham, MA, USA), a Halt Phosphatase Inhibitor Cocktail (Thermo Scientific, Waltham, MA, USA) along with 0.5 M EDTA and lysed for 20 min on ice. Then, the lysates were sonicated, centrifuged at 10,000 rpm for 10 min at 277 K, and the supernatants were collected for further analysis. The protein concentration was determined using a Micro BCA™ Protein Assay Kit (Thermo Scientific, Waltham, MA, USA) according to the manufacturer’s instructions. Equal amounts of the proteins (20 µg) were electrophoresed on SDS-Page gels and transferred onto nitrocellulose membranes. The membranes were blocked in 5% non-fat milk prepared in PBS containing 0.1% Tween-20 (TPBS) for 1 h. After blocking, the membranes were incubated with specific primary antibodies: p53, cyclin E1, heme oxygenase (HO-1), and GAPDH overnight at 277 K, then washed and incubated with horseradish peroxidase (HRP)-conjugated secondary antibodies for 1 h at room temperature. All of the antibodies were purchased from Cell Signaling (Danvers, MA, USA) and were diluted 1:1000 in 5% milk in TPBS. Finally, the membranes were washed and incubated with a SuperSignal™ West Pico Chemiluminescent Substrate (Thermo Scientific). The chemiluminescence signals were captured using a ChemiDoc™ XRS+ System (Bio-Rad, Hercules, CA, USA). The experiments were performed at least three times.

#### 2.4.7. Statistical Analysis

The results were expressed as the mean ± standard deviation (SD) from at least three independent experiments. Statistical analysis of the ROS measurements was performed using the two-tailed Student’s t-test. The statistical differences in the expression of proteins, the cell cycle’s progression, and Annexin V binding assay were calculated using the one-way ANOVA with a Bonferroni *post-hoc* test. A p-value of 0.05 or less was considered to be statistically significant. GraphPad Prism v.7.0 software (GraphPad Software, San Diego, CA, USA) was used for the analysis.

## 3. Results

### 3.1. Structural Analysis—XRD and Raman Studies

The X-ray diffraction pattern of the initial silver-silica nanocomposite showed a combination of the signal that originated from the crystalline and amorphous phases. The main diffraction lines indicated silver oxide (Ag_2_O) with a cubic structure (Pn-3m). Less intense peaks were found that were derived from β-Ag_2_CO_3_ with a hexagonal structure (P31c) and Ag_6_Si_2_O_7_ with a monoclinic structure (P2/n). The very wide hump appeared at the 2θ range 15–27^0^ correspond to amorphous silica (Figure 2). A similar diffraction pattern was identified in the Ag/SiO_2_@AS. The Rietveld refinement revealed comparable values for the same crystalline phases, and similar crystallite sizes were estimated for the most substantial peak for Ag_2_O in the <111>, <200>, and <220> directions (Table 1 and Table 2). Additionally, silver chloride (AgCl) with a cubic structure (Fm-3m) was also detected (Figure 2). In turn, the Ag/SiO_2_@CMC sample was characterized by the presence of the silver chloride with large crystallites, which was found as the main phase due to a sharp and robust diffraction pattern (Figure 2 and Table 2). The low intense diffraction peaks highlighted a nanocrystalline metallic silver (Ag^0^) with a cubic structure (Fm-3m). The absence of the diffraction peaks of Ag_2_CO_3_ and Ag_6_Si_2_O_7_ may have resulted from the detection limit or a very low amount of those phases in the composite.

The Raman spectrum of pure silica is usually divided into two regions of which the first (1) range includes bands of the medium-range order silica superstructures (200–800 cm^−1^) and the second (2) refers to the silicon-oxygen tetrahedral *Q^n^* (n = 0–4 and stands for the amount of bridging oxygen per SiO_4_ tetrahedron) modes (800–1250 cm^−1^) [56,57].

The band arrangement of the reference pure silica in the region (1) aligns with the spectra of the Ag/SiO_2_ and Ag/SiO_2_@AS, while the additional band that is located in the vicinity of 229 cm^−1^ correlates with an overlapping signal that originated from Ag-O vibration within the Ag_2_O group [44,58] and the Si-O-Si modes (Figure 3). Similar band positions found for the reference silica indicate a low chemical interaction of silver on a silica network [59,60,61]. The narrow bands between 350–470 cm^−1^ correspond to the n-membered rings (n > 5), the deformational modes of the silanol (Si–OH) group [57,62], or structural point defects (Figure 3). The most intense bands’ position and intensity are centered at 243, 271 cm^−1^ in the Ag/SiO_2_@CMC spectrum, suggest an atypical molecular configuration in which the chloride ligands probably surround the Ag^+^-ions octahedral coordination [63]. This molecular arrangement implies the activation of powerful stretching vibrations of the bridging Ag–Cl modes as well as the terminal chlorine atoms, respectively, for the lower and higher Raman frequencies (Figure 3). Similar to previous studies, the existence of unsaturated—Si–O* bonds favors the formation of the silanol group [57,62].

The bands of the region (2) are a marker for determining the quality of a three-dimensional SiO_2_ network. In this context, the Raman spectra of the Ag/SiO_2_ and Ag/SiO_2_@AS samples indicate a band arrangement similar to the band number and intensity, while their position differs slightly. Five bands around at 867, 914, 984, 1076, and 1146 cm^−1^ (reference Ag/SiO_2_ spectrum) and at 874, 939, 1009, 1085, and 1155 cm^−1^ (Ag/SiO_2_@AS) indicated the strong depolymerization of silica network correlating with the *Q*^0^, *Q*^1^, *Q^2^*, *Q*^3^, and *Q*^4^ units [64] of SiO_4_ tetrahedra (Figure 3) [56,65]. A low intense band about 800 cm^−1^ in both samples results from Si vibration trapped in an oxygen cage [56,64]. In turn, two low intense bands at 870 and 1061 cm^−1^ (Ag/SiO_2_@CMC) correspond to the *Q*^0^ and *Q*^3^/*Q*^4^ units, wherein its position and intensity relate to the reference silica and may suggest the low impact of AgCl clusters on depolymerization of the silica network. The bands above 1400 cm^−1^, which had relatively high linewidths, corresponded to the organic polymer residues that had been dropped into the colloidal suspension during the chemical synthesis.

### 3.2. Microscopic Observations vs the Chemical Composition

The SEM observations combined with the EDS analysis revealed the irregular size of the silica particles from (21.6 ± 1.2) µm to (497.4 ± 2.3) µm (gray areas in Figure 4) and the silver had dispersed with much more homogeneity on the silica carrier (brighter areas in Figure 4a–c). The TEM analysis showed that at the nanoscale, the silver was spread heterogeneously in the Ag/SiO_2_ within a silica matrix with an average particle size that was estimated based on a Gaussian function and that was approximated as d = (7.8 ± 0.3) nm (Figure 5a). The silver nanoparticles had spherical shapes for Ag/SiO_2_@CMC and were also more diversified and trended toward spheroidal- and irregular-shaped structures for Ag/SiO_2_@AS (Figure 5b,c). Hence, determining their size was difficult, and this resulted in estimating the approximate values as d = (44.1 ± 2.3) nm for Ag/SiO_2_@CMC and d = (6.6 ± 0.7) nm for Ag/SiO_2_@AS (Figure 5b,c). The diffraction pattern for all of the structures determined from the selected area electron diffraction (SAED) and XRD data coincided (Table 3). Silver was found in two valence states (Ag^+^ and Ag^2+^) in Ag/SiO_2_ and Ag/SiO_2_@AS, while in Ag/SiO_2_@CMC, it was in a metallic Ag^0^ or silver connected to chlorine. It is worth noting that the metallic and oxide silver structural parameters were similar to other silver-silica composites, which were presented in our earlier works [44,46].

The elemental distribution maps of Si, O, and Ag, as well as chlorine as an impurity that originated from the presence of additives in the organic polymers in the two modified silver-silica systems, were analyzed more in detail (Figure 4). According to these, the silver in the entire volume of the reference and sodium-alginate-modified composites was homogenously dispersed. In turn, it tended to agglomerate into a spherical shape in Ag/SiO_2_@CMC (Figure 4d–f). Moreover, the chlorine distribution map in the modified Ag/SiO_2_ compounds indicated a correlation between chlorine and silver due to the formation of the AgCl systems previously identified using the diffraction method (Figure 2). More precise information about elemental composition, chemical and electronic states considering the surface was obtained with X-ray photoemission spectroscopy (XPS) (Figure 6 and Figure 7). In this context, the O 1s line in the reference sample was deconvoluted into three components (530.35 eV, 531.3 eV, 533.1 eV), which were ascribed to the silver oxides. In the modified samples, the most intense components at binding energies 532.2 and 532.9 eV were assigned to SiO_2_, while the other states of oxygen denoted as SiO_2−*x*_, AgO, and O/Ag with peaks at 531.2, 530.9, and 529.3 eV. The Si 2p core lines revealed a non-stoichiometric SiO*_x_* (101.2 eV) and a stoichiometric SiO_2_ (103.2 eV) silica in each of the studied composites. Other peaks found at 100.2, 102.4, and 104.3 eV were correlated with the SiC (Ag/SiO_2_@AS), SiO_2−*x*_ (Ag/SiO_2_@CMC), and SiO/SiOH (Ag/SiO_2_) [66]. An interpretation of the Ag 3d lines indicated silver in its metallic state at the surface for all of the samples due to the photoelectron lines at the binding energies located in the interval from 368.3 to 368.6 eV (Figure 3). Other lines at 366.3 and 369.2 eV (Ag/SiO_2_) as well as at 366.1 eV (Ag/SiO_2_@CMC) were derived from the splitting of the 3d doublet equal 6.0 eV [67], was linked to the formation of the ionic silver cluster on the surface of the silver nanoparticles immediately after the synthesis [68]. The additional lines, which were located at 367.8 (Ag/SiO_2_@CMC) and 367.6 eV (Ag/SiO_2_@AS), resulted from the AgCl moieties and the formation of AgO [69], respectively. In turn, one of the components within the Cl 2p core level (Ag/SiO_2_@CMC) with binding energy equal to 198.4 eV was ascribed to the AgCl cluster, while the nature of the other components at 197.4 and 201.8 eV were derived from the chlorine ions and chlorine, which are covalently bonded with the carbon (Figure 7). When the bulk and surface chemical composition were compared, the macroscopically silver content was estimated as 12.6%, 6.8%, and 7.9%, while for the surface analysis, it was 1.6%, 0.6%, and 1.8%, respectively, for Ag/SiO_2,_ Ag/SiO_2_@CMC, and Ag/SiO_2_@AS. The chlorine content was 0.85% in Ag/SiO_2_@CMC, while during the bulk analysis, it was 3.4% and there was only a trace amount of 0.2% in Ag/SiO_2_@CMC and Ag/SiO_2_@AS, respectively (Table 4). It is worth noting that all of the studied nanocomposites’ surface was carbon contaminated and had an atomic concentration that ranged from 8.4% to 11.1% (Table 4). Moreover, as a result of the fitting analysis, the C 1s core-level spectra indicated C–C, C–O, C=O, and O-C=O bonds [70].

### 3.3. Antimicrobial Properties

The ‘spot test’ was used to determine the silver-silica nanocomposites’ antimicrobial properties and reference silver nitrate. The results are expressed as the minimum biocidal concentration (MBC), i.e., the lowest concentration (mg/L) that inhibited microbial growth (Figure 8, Table 5). The results showed that the silver-silica nanocomposites were able to inhibit microbial growth at the tested concentrations. The MBC values that were obtained for the reference Ag/SiO_2_ were ten-fold higher relative to the Ag^+^-ions for the Gram-negative bacteria and were at the same level for the Gram-positive strains. The results were similar for the Ag/SiO_2_@AS nanocomposite, except for *L. plantarum*, which had an MBC slightly higher than the initial sample. There were also some differences for Ag/SiO_2_@CMC for which the slightly lower silver content (6.8%/0.6%) translated into ten-fold lower toxicity for *P. aeruginosa* and *S. aureus* than other systems. Moreover, the toxicity pattern of the tested nanocomposites against yeast had lower antifungal properties but still ten-fold higher than the reference AgNO_3_ (Figure 8, Table 5).

### 3.4. Anticancer Studies

The reference and modified silver-silica nanocomposites were tested against different cancer cell lines: colon (HCT 116), breast (MCF-7), glioblastoma (U-251), and pancreatic (PANC-1), which represent the most common and aggressive tumors. The anticancer activity was evaluated in vitro based on an MTS assay (Table 6). According to this analysis, all of the investigated nanomaterials had excellent activity against all of the cancer cell lines. The highest activity was observed for MCF-7, which had an IC_50_ value of 11 mg/L for Ag/SiO_2_@AS. In general, there were no significant differences between the tested nanocomposites for these cell lines. However, more significant differences were detected for PANC-1, which also expressed a good susceptibility.

In the next step, the compound’s effect was tested based on the cell cycle and cell death induction. Here, there was a significant increase in the cell population in the G_2_/M phase for all compounds, which indicated the cell cycle inhibition in the phases above (Figure 9a). Furthermore, an experiment that enabled the type of cell death to be determined showed an increase of the apoptotic PANC-1 cell fraction after 48 h of cell treatment with all of the investigated nanocomposites. Here, the modified silver-silica nanocomposites had the most significant effect, i.e., ~50% increase in the number of apoptotic cells compared to the control. Further, a time-dependent kinetic experiment showed a growing trend in the ROS generation for the most cytotoxic agent (Ag/SiO_2_@CMC) (Figure 9c). An examination of the proteins using the Western Blot test confirmed the previous results. Here, several targets linked to the cell cycle inhibition, apoptosis, and ROS generation were investigated. It was found that Ag/SiO_2_@CMC caused the upregulation of the HO-1 protein without a significant impact on the p53 and cyclin E proteins.

## 4. Discussion

In recent years, much research has been focused on rapidly developing new kinds of nanomaterials with different target applications, e.g., photocatalytic or biological. These are usually dedicated to only one type of application. Therefore, the presented studies were focused on developing a new class of biologically active systems. This goal was achieved by (i) combining metal nanoparticles with an inorganic matrix or (ii) combining metal nanostructures with organic saccharide-derivatives (carboxymethylcellulose, sodium alginate, etc.). Despite the advantages and disadvantages of nanocomposites, the hybrid structures in the assumption ought to increase metallic silver stability. They also extend the effect of their action at a reduced concentration and intensify the entire system’s biological properties. Hence, the research focused on reducing the negative effects of the simple silver-silica system and its derivatives. To do this, the physicochemical studies were correlated to the impact of the newly synthesized materials on bacteria, fungi, and cancer cells in an attempt to explain the potential mechanism of cell death.

An inorganic silver-silica nanocomposite and two organic–inorganic systems that had been encapsulated by sodium alginate and carboxymethylcellulose were prepared. The SEM and TEM studies illustrated heterogeneously distributed spherical nanoparticles. The EDS and XPS highlighted that they were in the form of metallic silver and silver oxides. Their sizes were estimated to be less than 10 nm in the silica network (Ag/SiO_2_, Ag/SiO_2_@AS). In turn, spherical silver nanoparticles (Ag NPs) in the form of metallic silver and a silver-chlorine system ~44 nm in size were observed for Ag/SiO_2_@CMC. The silver concentration was estimated at a level of about 10 at.% (bulk) and 2 at.% (surface) for all of the systems (Table 4). The Raman investigations revealed that the synthesis procedure also influenced a structural modification within the silica network, which enabled interconnections to be formed between the positively charged silver and non-bonded oxygen atoms of the silica matrix. Also, the XRD studies showed that the Ag NPs were embedded in an amorphous silica network. These data are crucial in the context of antimicrobial and anticancer studies. Hence, the next part of the discussion focuses on the correlation between the physicochemical parameters and their impact on the biological systems.

One way to explain the antibacterial effect of silver-silica nanocomposites to look at a cell’s environmental condition, metabolism, or wall structure of a specific bacteria or yeast. During the data interpretation, crucial information might be that *L. plantarum* bacteria are aerotolerant organisms that prefer anaerobic conditions, while the other three tested bacteria are aerobic and anaerobic. The addition of external modifiers such as silver to their surroundings can generate oxidation stress in cells, which transforms an excess of oxygen and releases hydrogen peroxide, thus inducing the bacteria’s death [71]. Moreover, lower MBC values that are related to other strains can be explained through the synergistic interaction of metallic and oxidized (AgO, Ag_2_O) silver as in the Ag/SiO_2_ and Ag/SiO_2_@AS as well as metallic and clustered silver in the silver chloride configuration as in Ag/SiO_2_@CMC. In turn, the lowest MBC values that were found for Ag/SiO_2_ could have resulted from the slightly higher atomic concentration of silver in the reference sample relative to the other systems (Table 4). The other bacterial strains were 10- or 100-fold less affected on the silver-silica nanocomposites than on *L. plantarum* and the ionic reference (Table 5). At first glance, it appears that two tested Gram-positive bacteria interact differently on the Ag NPs, suggesting a different mechanism responsible for cell death in *S. aureus* than in the case of *L. plantarum*. According to Feng et al. [72], silver nanoparticles in the initial stage may affect the DNA molecules’ replication ability, while after the lengthy exposure may provide the cell wall breakdown and release the cell’s contents into the environment. Moreover, a long-term silver interaction could damage the recombinase A protein (recA) or affect the transcription, translation, and post-translational of the recA gene, which would actively reduce the possibility of damaged DNA being repaired. Similar antimicrobial effects were also reported for *E. coli* [72,73,74]. According to these reports, the similar MBC values that were obtained for *E. coli* and *S. aureus* led to the supposition that the cell death mechanism may be linked to silver ions that are released from the surface of the Ag NPs into the inside of bacteria and its interaction with the sulfur- and phosphorus-containing compounds. In turn, the wall structure differences between the Gram-positive and Gram-negative bacteria appear to have a negligible impact on cell death, despite the presence of lipopolysaccharide in the cell wall, which should help to trap and block the positive charges of silver. This is also contradictory with the literature data that illustrates that Gram-negative bacteria are less sensitive to Ag NPs than Gram-positive bacteria [75]. The impact of the particle shape and size on the cell membrane, which was estimated for the two systems (Ag/SiO_2_ and Ag/SiO_2_@AS) to be well below 10 nm and a mutual interaction of the metallic and oxidized forms of silver can also not be excluded. According to the literature, a similar combination of Ag NPs with spherical (20 nm) or irregular shapes (5–10 nm) disturbed the proper functioning of bacteria cells [76,77]. Other results were observed for Ag/SiO_2_@CMC, which mainly concerned Gram-positive (*S. aureus*) and Gram-negative (*P. aeruginosa*) bacteria. One possible explanation may be the low concentration of silver and its chemical trapping in the silver-chlorine system, limiting the release of silver ions into the environment (data not shown). Another hypothesis may correlate with a much higher silver particle size estimated as d = (44.1 ± 2.3) nm, while the most toxic objects were nanoparticles with sizes smaller than 10 nm [28,34,78]. Additionally, a lower concentration of silver or a larger average size of the nanoparticles could lower intracellular bioavailability of silver and weaker contact between the cell and the particle [79,80,81]. Hence, the antimicrobial effect may only be explained through metallic silver interaction with the outer microbial membrane, which influences a modification of its lipid layer. A different situation was found for the yeast. The low MBC values had probably resulted in the yeast’s specific cell wall composition, i.e., glucans, chitin, and mannoproteins, which causes rigidity in the overall wall structure and obstructs the possibility of connecting the silver particles to the surface, thereby reducing its positive impact.

Parallel to the nanocomposites’ silver-silica antimicrobial properties, the anticancer activity against four cancer cell lines was tested. Among these cell lines, colon and breast represent the most common cancers in the entire population, especially in developed countries. In turn, glioblastomas and pancreatic cancer are the most aggressive tumors and have a poor prognosis, in which the overall five-year survival rate does not exceed 5%. Unfortunately, determining the impact of silver or silver-based derivatives on human, normal, or cancer cells is problematic. One of the hypotheses reported by Hussain et al. suggests that the primary target sites of silver are the mitochondria, which are generally involved in maintaining the redox homeostasis in cells [82]. In this context, the cytotoxicity effect of silver may be associated with the size of the nanoparticles, i.e., 100 nm-sized objects might not be able to enter cells, while a 10 nm sized particle can easily penetrate the interior of cells, which is prone to a strong release of silver ions. As a result, smaller Ag NPs could impact (i) the generation of reactive oxygen species (ROS), (ii) glutathione (GSH) depletion, (iii) the upregulation of heme oxygenase (HO-1) expression, (iv) a decrease in the enzyme activity of the superoxide dismutase (SOD) or (v) a mitochondrial membrane disruption [16,17,18,19,83]. Other studies demonstrated that Ag NPs might activate the redox-sensitive gene products such as the p38 mitogen-activated protein kinase (MAPK), nuclear factor E2-related factor-2 (Nrf2), and nuclear factor-kappa b (NF-κB) [84]. The ability of Ag NPs to induce an inflammatory response of the immune cells in vitro was also recently reported [34].

According to the experimental data, although the highest resistance was detected for the glioblastoma cell line, this type of cancer often exhibits drug-resistance. In turn, the lowest IC_50_ values were detected for MCF-7 and PANC-1. However, there were more significant differences between the tested nanocomposites for pancreatic cancer. Therefore, the PANC-1 cell line was selected for further studies, while the physicochemical data may help to understand the molecular mechanism of action of the silver-silica composite and its derivatives concerning this cancer cell line.

First of all, it is worth noting that both the PANC-1 and U-251 cell lines have a missense point mutation in the *TP53* gene, which causes amino acid to change an arginine to histidine in codon 273 [85,86]. This mutation is responsible for increasing the oncogenic potential, the deregulation of the cell cycle, and the ability to metastasize [87]. To look more deeply at the problem mentioned above and to attempt to explain the cytotoxic effect of the synthesized silver-silica nanocomposites, a cell cycle inhibition test was performed on the selected PANC-1 model. The cell cycle studies revealed a checkpoint regulation and cell cycle inhibition in the G_2_/M phase in all of the studied systems (Figure 9a). According to the literature data, cell cycle arrest in the S and G_2_/M phases might correspond to silver nanoparticles’ presence [17,19,21,22]. In contrast, the differences in a silica carrier structure seem to be less critical or even negligible. In this context, silver nanoparticles might cause morphological changes due to their interactions with cell permeability, ROS production, or an enzyme system alteration [88]. Ag NPs can also modulate the gene and protein expression, which are responsible for DNA repair [20], impact Chk1 kinase, prevent cdc25C activation, and repair DNA damage [89]. In turn, the effect of DNA damage may have resulted from direct interaction with the p53/p21 signaling pathway, which is responsible for the G_2_/M transition control system [90].

To shed more light on the type of cell death, the further possible pathways of the cell cycle inhibition were examined. The cells’ exposure to the silver-silica systems for 48 h caused an increase in the apoptotic PANC-1 cell fraction (Figure 9b). More precisely, the number of apoptotic cells increased only slightly (~15%) relative to the reference Ag/SiO_2_, while for other systems, it was even more than 50% compared to the control. Unexpectedly, this effect was negatively correlated with the silver concentration in the system (Table 4). Hence, it is worth looking at the other factors that might explain the studied systems’ cytotoxicity effect. One hypothesis might be the metallic and oxidized silver nanoparticles’ synergistic effect as detected for Ag/SiO_2_ and Ag/SiO_2_@AS. In those systems, Ag NPs are smaller than 10 nm, enabling them to easily enter the interior of cells and promote the release of silver ions into the environment. Similar results were reported by Zielinska et al. [91] or Cameron et al. [83]. In turn, the Ag/SiO_2_@CMC system’s high anticancer effect, which was characterized by a particle size that was larger than 44 nm, could result from a slightly different mechanism of action. Here, more than 60% apoptotic or dead cells were observed (Figure 9b). The physicochemical interpretation indicated silver chloride and/or a metallic silver core with chloride, metallic, or clustered ions around the core, which can be an important factor that should be considered. Here, one can assume that the synergistic effect of the ions that were released from the surface of AgCl might correlate with the cytotoxicity data that was obtained for small-sized systems. Another hypothesis might be the impact of neutrally charged metallic silver, which can penetrate the lipid membrane cell and release the ions inside the cell. However, the molecular mechanisms of cancer cell death are still not fully understood and remain subject to further discussions.

One of the potential DNA damage factors is the generation of reactive oxygen species (ROS) [16,19,92,93]. The total number of ROS in a time-dependent kinetic experiment was estimated for the most cytotoxic Ag/SiO_2_@CMC system (Figure 9c) to verify this hypothesis. The growing trend in the number of ROS that was observed in the first 6 h and then up to 24 h correlates with the release of the redox-active ions and enforces the cellular pathways that were postulated above. Hence, the ROS data can clarify the response to G_2_/M arrest, especially by looking more precisely at the alterations in the G_2_ checkpoint-associated proteins’ levels in silver-silica and its derivatives-treated cells. In this context, it is crucial to consider that the checkpoints G_1_/S and the G_2_/M may correlate with the impact of protein *p*53, which is known as the “guardian of the genome”. It is also usually engaged in cellular stress, DNA-damage, or checkpoint failure [94].

On the other hand, mutations in the gene encoding *p*53 are found in half of all cancers. Therefore, to look more deeply at the potential mechanism of silver action, a Western Blot test was performed exemplary on Ag/SiO_2_@CMC. The obtained data confirmed previous results about cell cycle inhibition, apoptosis, or ROS generation hypothesis. Furthermore, immunoblot analyses revealed the upregulation of the stress-induced heme oxygenase (HO-1) protein and the lack of any significant influence on the *p*53 protein, which is usually mutated in cancer cells, thus suggesting a *p*53-independent mechanism of action. This data point to the induction of the oxidative stress in cells due to the presence of silver-silica nanocomposites and suggest that a mechanism of action has resulted from the induction of radical AS (Figure 9c). Additionally, another confirmation of the G_2_/M arrest may also be the data that illustrated no cyclin E expression changes that involve the G_1_ phase of the cell cycle (Figure 9d). Moreover, our results seem to confirm the previous postulates of Eom et al., who reported that silver nanoparticles could affect the oxidative stress pathway—p38 MAPK via ROS formation, which then induces DNA damage and cell cycle arrest [84].

## 5. Conclusions

Chemical synthesis was used to prepare the silver-silica systems and their derivatives, which were fabricated by modifying the reference silver-silica system by sodium alginate and carboxymethylcellulose. The silver nanoparticles in the Ag/SiO_2_ and Ag/SiO_2_@AS were distributed heterogeneity within the silica carrier, whereby the silver occurred in different states: oxides, metallic or ionic. The silver-silica system, which was chemically modified by carboxymethylcellulose, was characterized by larger silver nanoparticles (~44 nm) that tended to strongly agglomerate. Chemically, this system was composed of metallic silver and silver that was clustered within the silver-chlorine system. The silver concentration was estimated at a level of about 10 at.% (bulk) and 2 at.% (surface) for all of the systems, which resulted in an antimicrobial and anticancer effect. It was found that the tiny silver nanoparticles that were found in Ag/SiO_2_ and Ag/SiO_2_@AS influenced the cell growth inhibition, especially in *E. coli*, *S. aureus,* and *L. Plantarum*, while the massive silver nanoparticles in Ag/SiO_2_@CMC had a slightly lower antimicrobial effect. A similar cytotoxicity effect on the cancer cells was found for the physicochemical properties independently. The more detailed studies that were performed on the example of Ag/SiO_2_@CMC illustrated that the anticancer effect resulted from the reactive oxygen species generation.

## Figures and Tables

**Figure 1 nanomaterials-10-02551-f001:**
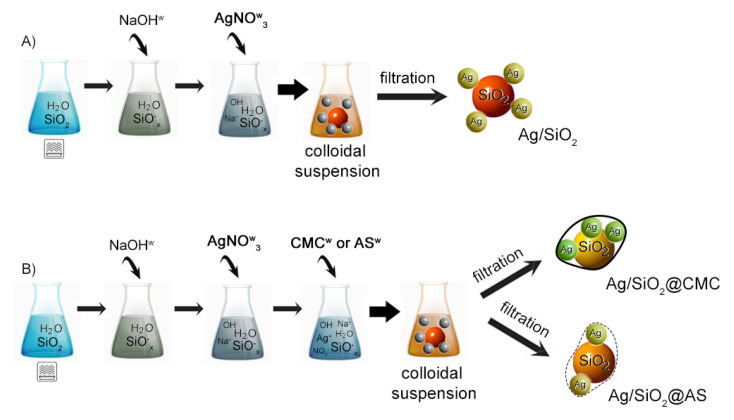
Two synthesis routes (**A**,**B**) were considered to fabricate the powder (**A**) silver-silica nanocomposite and (**B**) its derivatives.

**Figure 2 nanomaterials-10-02551-f002:**
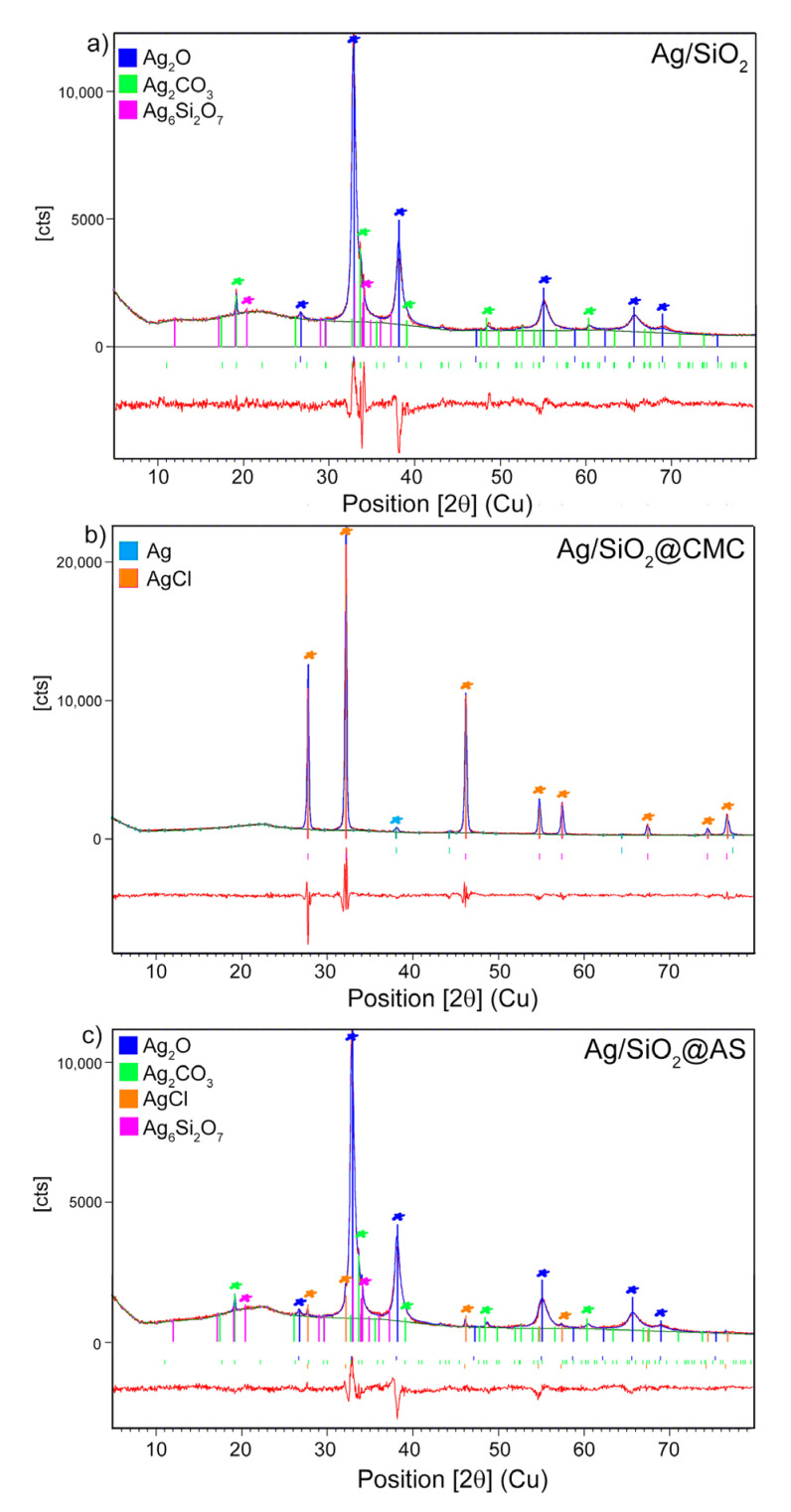
XRD pattern of (**a**) initial Ag/SiO_2_ powder and modified systems by (**b**) carboxymethylcellulose: Ag/SiO_2_@CMC and (**c**) sodium alginate: Ag/SiO_2_@AS with the Rietveld refinement (red line).

**Figure 3 nanomaterials-10-02551-f003:**
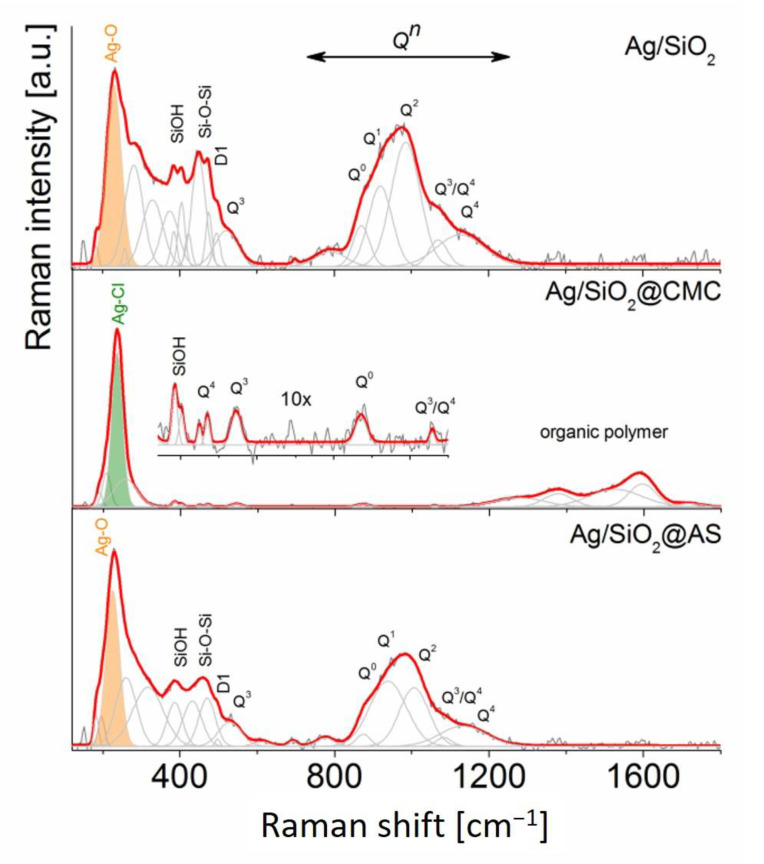
The Raman spectra of the silver-silica (Ag/SiO_2_) and modified nanocomposites through carboxymethylcellulose (Ag/SiO_2_@CMC) and sodium alginate (Ag/SiO_2_@AS) were in the 120–1800 cm ^−1^ region. Additionally, to better illustrate the silica matrix signal, the 350–1100 cm^−1^ region in Ag/SiO_2_@CMC was magnified 10×. Each spectrum was fitted using the Voigt function and a minimum number of components.

**Figure 4 nanomaterials-10-02551-f004:**
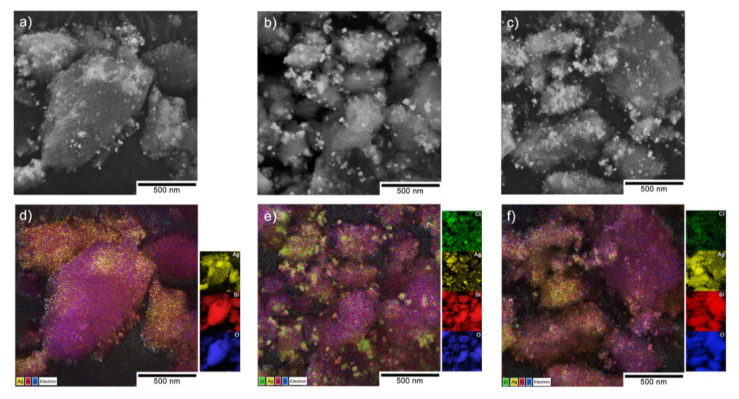
Backscattered electron (BSE)-SEM (**a**–**c**) images present the morphology of (**a**) Ag/SiO_2_, (**b**) Ag/SiO_2_@CMC, (**c**) Ag/SiO_2_@AS and the SEM-energy dispersive spectrometer (EDS) chemical composition maps of the (**d**) Ag/SiO_2_, (**e**) Ag/SiO_2_@CMC, and (**f**) Ag/SiO_2_@AS nanocomposites and their derivatives. The small insets highlight the individual elemental distribution maps of Ag (yellow), Si (red), O (blue), and Cl (green) from the EDS. CMC means carboxymethylcellulose, while AS refers to sodium alginate.

**Figure 5 nanomaterials-10-02551-f005:**
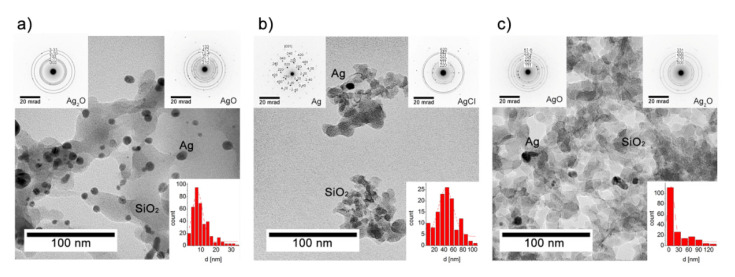
High resolution transmision electron microscopy (HRTEM) images of (**a**) the pure silver-silica nanocomposite and the chemically modified silver-silica derivatives; (**b**) Ag/SiO_2_@CMC and (**c**) Ag/SiO_2_@AS with an inset showing the selected area electron diffraction (SAED) patterns for Ag and the size distribution of Ag fitted by the Gaussian function.

**Figure 6 nanomaterials-10-02551-f006:**
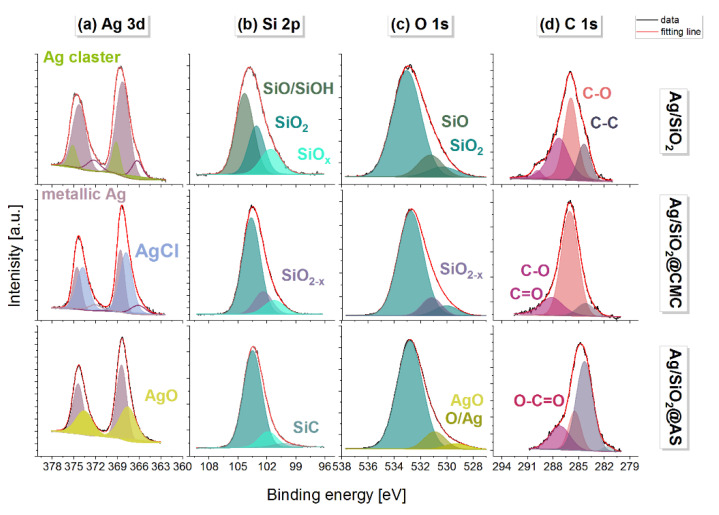
The XPS spectra of (**a**) Ag 3d, (**b**) Si 2p, (**c**) O 1s, and (**d**) the C 1s core levels with the fitting result. The individual colors on the specific lines’ core levels refer to the individual chemical surroundings linked to the silver, silica, oxygen, and carbon.

**Figure 7 nanomaterials-10-02551-f007:**
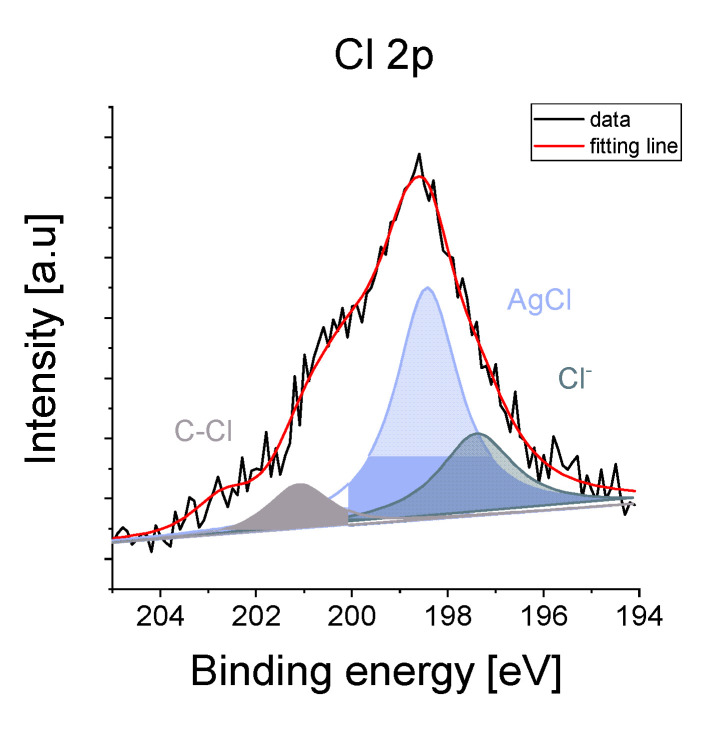
The XPS spectrum of the Cl 2p core level with the result of the fitting for Ag/SiO_2_@CMC.

**Figure 8 nanomaterials-10-02551-f008:**
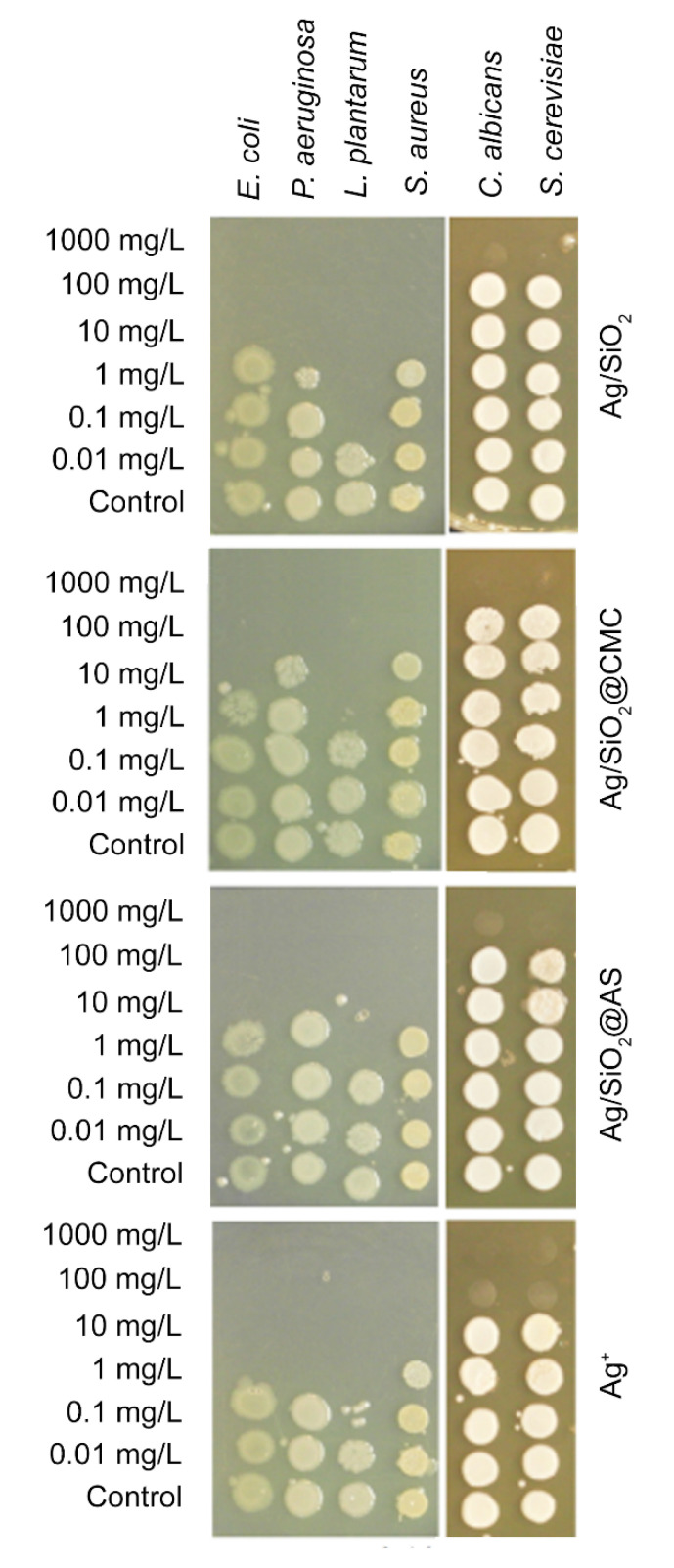
The ability of the Gram-negative (*Escherichia coli, Pseudomonas aeruginosa*), Gram-positive bacteria (*Lactobacillus plantarum*, *Staphylococcus aureus)* and yeast strains (*Candida albicans*, *Saccharomyces cerevisiae*) to form a colony after a 24 h exposure to the Ag/SiO_2_, Ag/SiO_2_@CMC, Ag/SiO_2_@AS, and Ag^+^-ions (reference). After exposure, the cells (5 µL) were transferred onto a toxicant-free agarized growth medium. The lowest concentration (mg/L), providing total microbial growth inhibition was considered minimum biocidal concentration (MBC).

**Figure 9 nanomaterials-10-02551-f009:**
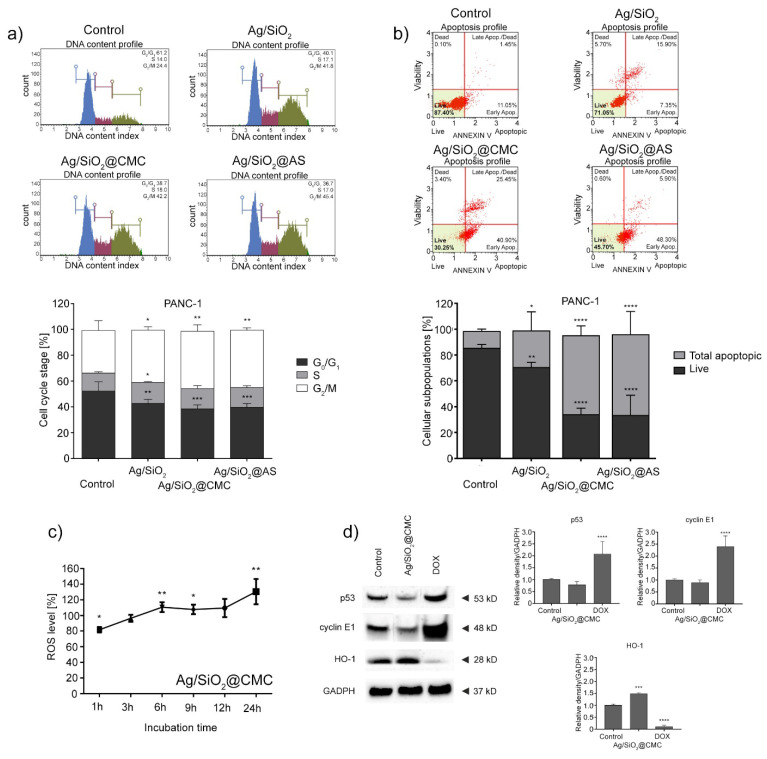
Impact of the tested nanocomposites at a 30 mg/L concentration on regulating the cell cycle (**a**) and inducing apoptosis (**b**) in the PANC-1 cells. Effect of the tested Ag/SiO_2_@CMC nanocomposite on the level of reactive oxygen species (ROS) in the PANC-1 cells. Data normalized to the untreated cells (control) (**c**). Impact of Ag/SiO_2_@CMC on the expression of the p53, cyclin E1, and HO-1 proteins in the PANC-1 cells. The densitometric analysis of these proteins was normalized to GADPH (**d**). The results from all experiments are shown as the mean ± standard deviation (SD) of three independent measurements. Any statistical differences from the cell cycle, apoptosis, and immunoblotting experiments were analyzed using a one-way ANOVA with Bonferroni’s *post-hoc* test. Data from ROS measurements were analyzed using the Student’s *t*-test. Statistical significance: * *p* < 0.05, ** *p* < 0.01, *** *p* < 0.001, **** *p* < 0.0001 compared to the control group.

**Table 1 nanomaterials-10-02551-t001:** The lattice parameters were determined from the XRD data and the analysis of the Rietveld refinement.

Phase (Space Group)	Ag (Fm-3 m)	AgCl (Fm-3 m)	Ag_2_O (Pn-3 m)	β-Ag_2_CO_3_ (P3_1_c)	
Lattice parameters	a_0_ [Å]	a_0_ [Å]	a_0_ [Å]	a_0_ [Å]	c_0_ [Å]
Ag/SiO_2_	-	-	4.710(9)	9.208(4)	6.490(7)
Ag/SiO_2_@CMC	4.085(5)	5.550(6)	-	-	-
Ag/SiO_2_@AS	-	5.554(9)	4.714(4)	9.202(2)	6.485(2)

**Table 2 nanomaterials-10-02551-t002:** The crystallite sizes were estimated based on the XRD data.

Ag_2_O (AgSiO_2_)		AgCl (AgSiO_2_@CMC)		Ag_2_O (AgSiO_2_@AS)	
Direction	Crystallite Size [Å]	Direction	Crystallite Size [Å]	Direction	Crystallite Size [Å]
<111>	172	<111>	901	<111>	168
<200>	135	<200>	747	<200>	132
<220>	82	<220>	674	<220>	81

**Table 3 nanomaterials-10-02551-t003:** Crystallographic data (lattice parameters, system, and space group) of the initial and modified silver-silica nanocomposites that were obtained based on the TEM + SAED data.

Element	Ag/SiO_2_		Ag/SiO_2_@CMC		Ag/SiO_2_@AS	
	AgO	Ag_2_O	Ag	AgCl	AgO	Ag_2_O
a_0_ [Å]	5.8517	3.072	4.0855	5.5463	5.8517	3.072
b_0_ [Å]	3.4674	3.072	4.0855	5.5463	3.4674	3.072
c_0_ [Å]	5.4838	4.941	4.0855	5.5463	5.4838	4.941
system	monoclinic	trigonal	cubic	cubic	monoclinic	trigonal
space group	P21/c	P-3m1	Fm-3m	Fm-3m	P21/c	P-3m1

**Table 4 nanomaterials-10-02551-t004:** The average atomic element concentration for the initial and modified silver-silica nanocomposites was estimated for the bulk (SEM-EDS) and surface (XPS). The data in the parentheses refer to the standard deviation data that was obtained based on the measurements of five different points.

El.	EDS	XPS	EDS	XPS	EDS	XPS
	Ag/SiO_2_ [at.%]		Ag/SiO_2_@CMC [at.%]		Ag/SiO_2_@AS [at.%]	
O	63.3 (3.0)	59.1	69.6 (1.6)	59.4	71.1 (0.7)	58.1
Si	24.1 (1.8)	28.8	20.1 (0.5)	30.7	20.9 (0.9)	28.9
Cl	-	-	3.4 (0.9)	0.9	0.2 (0.1)	-
Ag	12.6 (2.2)	1.6	6.8 (1.1)	0.6	7.9 (1.3)	1.8
C	-	10.5	-	8.4	-	11.2

**Table 5 nanomaterials-10-02551-t005:** Minimum biocidal concentration (MBC) for Ag/SiO_2_ and its derivatives after a 24 h exposure in deionized water at 297 K on the bacterial and yeast strains in nominal concentrations from 0.01 up to 1000 mg/L. The results were combined with the reference Ag^+^-ions (AgNO_3_). Different colors indicate different MBC values. All of the values were obtained based on experiments in three replicates.

Nanocomposites	Gram-Negative Bacteria		Gram-Positive Bacteria		Yeast	
	*E. coli*	*P. aeruginosa*	*L. plantarum*	*S. aureus*	*C. albicans*	*S. cerevisiae*
AgSiO_2_	10	10	0.1	10	1000	1000
AgSiO_2_@CMC	10	100	1	100	1000	1000
AgSiO_2_@AS	10	10	1	10	1000	1000
AgNO_3_	1	1	0.1	10	100	100

**Table 6 nanomaterials-10-02551-t006:** Anti-proliferative activity (IC_50_ values) of the tested nanocomposites against different human cancer cells.

Nanocomposite	IC_50_ Values [mg/L]			
	HCT116	MCF-7	U-251	PANC-1
Ag/SiO_2_	24.0 ± 9.3	13.7 ± 1.6	34.2 ± 8.5	21.0 ± 2
Ag/SiO_2_@CMC	24.2 ± 2.4	16.1 ± 4.7	38.7 ± 4.3	12.6 ± 6.0
Ag/SiO_2_@AS	24.0 ± 6.3	11.1 ± 2.0	30.1 ± 5.2	19.7 ± 7.5

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
