# Peer review of "An Organic–Inorganic Hybrid Nanocomposite as a Potential New Biological Agent"

_nanomaterials, 2020, doi:10.3390/nano10122551_

Round 1

Reviewer 1 Report

REVIEW OF MANUSCRIPT  (ISSN 2079-4991)

The manuscript “Organic-inorganic hybrid nanocomposite as a potential new biological agent” describes interestingly silica-based nanocomposites and their derivatives functionalized by silver ions or nanoparticles. Each of the systems has biologically tested, considering their impact on Gram-positive and Gram-negative as well as cytotoxicity effects on a panel of cancer cell lines, including colon, breast, pancreatic cancers, and glioblastoma. According to antimicrobial data, tiny silver nanoparticles implicated cell growth inhibition. Furthermore, it has been found that Ag/SiO2@CMC implied the generation of reactive oxygen species (ROS) as well as the induction of apoptosis by the p53-independent mechanism. The paper shows an interesting study but format and scientific quality of the paper should be improved. In this scenario, I strongly recommend publish after major revisions.

Abstract

English improvement is recommended. The next sentence has a weird verbal tense: Each of the systems has biologically tested.

The abstract has too long sentences.

Reading the abstract it is not clear if the silica nanoparticles are functionalized with silver ions or with silver ions or silver nanoparticles.

I would recommend a revision of the abstract for the better understanding.

Introduction

Line 55: The author should use capital letter.

Line 77: The author should remove the underlined.

Experimental Part

Chemicals section is recommended.

Line 136: The begging of the sentence is not clear.

Synthesis paragraph is not clear. The lector can not reproducible the method that authors are explained.

The synthesis of silver-silica nanocomposite and its derivatives should be explained at the Discussion section, no at the Experimental part.

Line 286, line 306: The author should decrease the letters size. They are too big.

Results

The Figure 2 should increase the quality, its difficult to see the diffraction lines.

Table 5 is missed.

Figure 8 is difficult to understand.

Table 6 is part of the Viability tests and shows The antimicrobial test.

Discussion

Discussion should be improved.

Author Response

The manuscript “Organic-inorganic hybrid nanocomposite as a potential new biological agent” describes interestingly silica-based nanocomposites and their derivatives functionalized by silver ions or nanoparticles. Each of the systems has biologically tested, considering their impact on Gram-positive and Gram-negative as well as cytotoxicity effects on a panel of cancer cell lines, including colon, breast, pancreatic cancers, and glioblastoma. According to antimicrobial data, tiny silver nanoparticles implicated cell growth inhibition. Furthermore, it has been found that Ag/SiO2@CMC implied the generation of reactive oxygen species (ROS) as well as the induction of apoptosis by the p53-independent mechanism. The paper shows an interesting study but format and scientific quality of the paper should be improved. In this scenario, I strongly recommend publish after major revisions.

Abstract

English improvement is recommended. The next sentence has a weird verbal tense: Each of the systems has biologically tested.

The abstract has too long sentences.

Reading the abstract it is not clear if the silica nanoparticles are functionalized with silver ions or with silver ions or silver nanoparticles.

I would recommend a revision of the abstract for the better understanding.

The abstract has been rewritten and English corrected.  

Introduction

Line 55: The author should use capital letter.

Line 77: The author should remove the underlined.

It has been changed.

Experimental Part

Chemicals section is recommended.

Line 136: The begging of the sentence is not clear.

Synthesis paragraph is not clear. The lector can not reproducible the method that authors are explained.

The synthesis of silver-silica nanocomposite and its derivatives should be explained at the Discussion section, no at the Experimental part. –

Line 286, line 306: The author should decrease the letters size. They are too big.

We changed the name of this part name like it is in the template. Therefore, we think it will be more reasonable to show the manuscript scheme proposed previously and easy to understand the synthesis problem.

Results

The Figure 2 should increase the quality, its difficult to see the diffraction lines.

I hope we improved the quality of Figure 2.

Table 5 is missed.

You have right, in the paper, it was a problem with Tables' labeling. Now we hope everything should be labeled well.

Figure 8 is difficult to understand.

I hope we improved Figure 8, especially in the context of its readability.

Table 6 is part of the Viability tests and shows The antimicrobial test.

You have right, in the paper, missed Table 5 and this misunderstanding came from this reason. Now we hope everything should work well.

Discussion

Discussion should be improved.

We have been improved the Discussion part. We have rewritten and added some fragments and corrected English.

Reviewer 2 Report

The manuscript entitled "Organic-inorganic hybrid nanocomposite as a potential new biological agent" by Dulski and coworkers describes the preparation of silver-silica systems and their derivatives with sodium alginate and carboxymethylcellulose. The prepared systems have been characterized from structural and morphological point of view, and were evaluated their antimicrobial properties and anticancer activity. The work reported is interesting and could be considered for publication after major revision.

Following are some questions/recommendations/suggestions:

  • Page 2, line 72:The sentence is underlined.
  • Page 3, line 127: please revise the sentence "The electronic structure and oxidation state of silver were determined using and X-ray photoelectron spectroscopy (XPS)."
  • Page 3, line 140: What is the reason of writing words "carboxymethylcellulose", "sodium chloride" and "sodium glycolate" hyperlinked?
  • The authors affirm: "Then, the water dissolved sodium hydroxide and water solution of the silver nitrate were added dropwise to the silica suspension, continuously stirred on a magnetic stirrer to fabricate the silver-silica colloid." How long the mixture has been stirred? At which temperature?
  • In the figure 8, at page 14, how the authors will explain the spots appearance in the images in line with 1000 mg/L?
  • The percent loading of Ag2O and polymers should be given. Did the authors perform TGA and nitrogen sorption measurements? SEM-EDS images provide only information on atom distribution.
  • Did the authors compare their results with the results reported by Rangelova et al. ("Preparation and characterization of SiO2/CMC/Ag hybrids with antibacterial properties",Carbohydrate Polymers,Vol.101, 2014,Pages 1166-1175). There is a similar study.

Author Response

The manuscript entitled "Organic-inorganic hybrid nanocomposite as a potential new biological agent" by Dulski and coworkers describes the preparation of silver-silica systems and their derivatives with sodium alginate and carboxymethylcellulose. The prepared systems have been characterized from structural and morphological point of view, and were evaluated their antimicrobial properties and anticancer activity. The work reported is interesting and could be considered for publication after major revision.

Following are some questions/recommendations/suggestions:

  • Page 2, line 72: The sentence is underlined.
  • Page 3, line 127: please revise the sentence "The electronic structure and oxidation state of silver were determined using and X-ray photoelectron spectroscopy (XPS)."
  • Page 3, line 140: What is the reason of writing words "carboxymethylcellulose", "sodium chloride" and "sodium glycolate" hyperlinked?

It has been changed. 

  • The authors affirm: "Then, the water dissolved sodium hydroxide and water solution of the silver nitrate were added dropwise to the silica suspension, continuously stirred on a magnetic stirrer to fabricate the silver-silica colloid." How long the mixture has been stirred? At which temperature?

We changed the synthesis part. It should be easiest to understand. As you find in this part, the suspension has been stirred for 1h. The temperature conditions had not been applied.

  • In the figure 8, at page 14, how the authors will explain the spots appearance in the images in line with 1000 mg/L?

You have absolutely right in the case of such a Figure. Unfortunately,  this graphical problem has appeared during Figure preparation. In the newest version, we checked our laboratory notes and the Figure was corrected according to this data.

  • The percent loading of Ag2O and polymers should be given. Did the authors perform TGA and nitrogen sorption measurements? SEM-EDS images provide only information on atom distribution.

The concentration of silver and polymer were added into the synthesis part. Unfortunately, TGA was not considered in the paper, but you have right. This analysis can be useful in the context of the thermal stability of the system. We want to perform TGA in the future, for sure. A similar situation was in the case of the BET analysis. However, here, an additional problem appears i.e. we don't have access to the appropriate device. Similar to previously, it is a good point and BET analysis is worth taking into consideration during future studies, especially in the context of silica and porous composites.

  • Did the authors compare their results with the results reported by Rangelova et al. ("Preparation and characterization of SiO2/CMC/Ag hybrids with antibacterial properties",Carbohydrate Polymers,Vol.101, 2014,Pages 1166-1175).

The above manuscript is interesting. We want to emphasize that the authors used different precursors to nanocomposite fabrication that may provide different results, especially in the bacterial effect of SiO2/Ag/CMC. However, some interesting observations and data were suggested and therefore, our paper has been supplemented by this paper. Thank you for taking our attention to the paper.

Reviewer 3 Report

Article: Organic-inorganic hybrid nanocomposite as a potential new biological agent

Authors: Dulski, K. Malarz, M. Kuczak, K. Dudek, K. Matus, S. Sułowicz, A. Mrozek-Wilczkiewicz and A. Nowak

The study presented a mixture of environmental, and biological factors, that implicates the development of modern civilization diseases. The authors prepared silver-silica systems and their derivatives fabricated due to its modification by sodium alginate and carboxymethylcellulose.According to this approach, two different molecular nanostructures have been achieved: on system with a silver chlorine salt immersed on a silica carrier with coagulated particles with size d= (44.1±2.3) nm coexisting with metallic silver, as well as second with synergistically interacted metallic and oxidized silver nanoparticles with average size equal d= (6.6±0.7)nm spread on a structurally defected silica network. Each of the systems has biologically tested, considering their impact on Gram-positive and Gram-negative as well as cytotoxicity effects on a panel of cancer cell lines, including colon, breast, pancreatic cancers and glioblastoma. They want to develop a new class of biologically active agents. They realized two folding: combining a) metal nanoparticles with inorganic matrix and b) metal nanostructures with organic saccharide-derivatives (carboxymethylcellulose, sodium alginate). They found out that both solutions have some advantages and disadvantages, so they focused on an attempt to reduce negative aspects and fabricate synergistically interacting organic-inorganic structure. I would like to recommend this article for publication. However, I would like to mention few comments. They should be treated as minor:

Take care of spaces between words and/or symbols/references in the whole text: line 46, 52, 70, 87, 92, 113

Change:

  • Line 189: ◎m→ nm
  • Line 191: ◎m→ nm

Author Response

The study presented a mixture of environmental, and biological factors, that implicates the development of modern civilization diseases. The authors prepared silver-silica systems and their derivatives fabricated due to its modification by sodium alginate and carboxymethylcellulose.According to this approach, two different molecular nanostructures have been achieved: on system with a silver chlorine salt immersed on a silica carrier with coagulated particles with size d= (44.1±2.3) nm coexisting with metallic silver, as well as second with synergistically interacted metallic and oxidized silver nanoparticles with average size equal d= (6.6±0.7)nm spread on a structurally defected silica network. Each of the systems has biologically tested, considering their impact on Gram-positive and Gram-negative as well as cytotoxicity effects on a panel of cancer cell lines, including colon, breast, pancreatic cancers and glioblastoma. They want to develop a new class of biologically active agents. They realized two folding: combining a) metal nanoparticles with inorganic matrix and b) metal nanostructures with organic saccharide-derivatives (carboxymethylcellulose, sodium alginate). They found out that both solutions have some advantages and disadvantages, so they focused on an attempt to reduce negative aspects and fabricate synergistically interacting organic-inorganic structure. I would like to recommend this article for publication. However, I would like to mention few comments. They should be treated as minor:

Take care of spaces between words and/or symbols/references in the whole text: line 46, 52, 70, 87, 92, 113

Change:

  • Line 189: ◎m→ nm
  • Line 191: ◎m→ nm

All editing mistakes have been corrected.

Round 2

Reviewer 1 Report

The paper has improved the quality, the I accept the Manuscript.

Reviewer 2 Report

The authors answers are satisfactory. The manuscript has been improved and could be considered for publication in the Journal Nanomaterials.